# Offline Reinforcement Learning with Bayesian Flow Networks

## Abstract

This paper presents a novel approach to reinforcement learning (RL) utilizing Bayesian flow networks for sequence generation, enabling effective planning in both discrete and continuous domains by conditioning on returns and current states. We explore two conditioning strategies: state inpainting and a classifier-free method. Experimental results demonstrate the robustness of our method across various environments. It navigated gridworld environments in discrete settings, without sacrificing performance in continuous tasks compared to current state of the art . The results highlight our approach's ability to effectively capture spatial and temporal dependencies through a specialized neural network architecture combining 2D convolutions with a temporal u-net.

## 1 Introduction

Offline reinforcement learning (RL), also known as batch RL, is a powerful paradigm that leverages previously collected data to learn effective policies. Unlike online RL, where agents interact directly with an environment, offline RL operates in a safer mode, utilizing historical data without risking real-time exploration. This safety advantage is particularly crucial in domains like autonomous driving and medical applications, where explorative policy collection can be hazardous. By drawing from pre-existing datasets, offline RL enables more efficient learning, making it a promising approach for real-world applications where data collection can be costly, time-consuming, or impractical. In recent years, there has been a surge of interest in offline RL due to its potential to address the challenges of sample inefficiency and exploration in traditional RL settings. However, offline RL also poses unique challenges, such as distributional shift and data quality issues (Agarwal et al., 2020; Levine et al., 2020).

Recent advancements in conditional generative modeling offer an alternative approach to traditional offline RL methods. Sequence modeling, in particular, has gained prominence (Janner et al., 2021; Ajay et al., 2023; Chen et al., 2021). By viewing RL as a sequence modeling problem, we can leverage the power of generative models to learn effective policies from already collected datasets. This perspective offers several advantages, such as the ability to capture temporal dependencies and complex interactions within the data. Moreover, conditional generative models allow for the generation of counterfactual trajectories, enabling robust policy evaluation and exploration of alternative decision-making strategies conditioned on return, desired goal state, or other desired behaviour. However, handling high-dimensional action or state spaces can be computationally intensive and may necessitate innovative approaches to maintain tractability. Despite this challenge, the integration of conditional generative modeling and RL holds promise for addressing the limitations of traditional methods and advancing the state-of-the-art in offline reinforcement learning.

## 2 Preliminaries

This section introduces all the necessary background to follow the related works and method sections.

## 2.1 Reinforcement Learning

Reinforcement learning (RL) is a framework for learning to make decisions in an environment (Sutton & Barto, 2018). The interactions with the environment are modelled as a Markov decision process (MDP), which is a tuple $(\mathcal{S}, \mathcal{A}, \mathcal{P}, \mathcal{R}, \gamma)$, where $\mathcal{S}$ is the state space, $\mathcal{A}$ is the action space, $\mathcal{P}$ is the transition function, $\mathcal{R}$ is the reward function, and $\gamma$ is the discount factor. In an environment with state $s$, the next state, $s' \sim \mathcal{P}(s, a)$, is only dependent on the current state and action, not the history of previous states and actions. In other words, it has the Markov property. The goal of RL is to learn a policy $\pi : \mathcal{S} \rightarrow \mathcal{A}$ that maximizes the expected return $\mathbb{E}[R_t]$, where $R_t = \sum_{i=0}^{\infty} \gamma^i r_{t+i}$, is the discounted cumulative reward and $r_t$ is the reward recieved at time $t$.

The exploration-exploitation trade-off is a fundamental challenge in reinforcement learning, typically associated with online learning scenarios where agents iteratively interact with an environment to learn optimal policies. Exploration involves sampling actions to gather information about the environment, potentially leading to the discovery of better strategies, while exploitation entails leveraging known information to maximize immediate rewards. Much of the research in online RL is dedicated to striking a balance between exploration and exploitation, devising algorithms that effectively navigate this trade-off to converge to optimal or near-optimal policies.

## 2.2 Offline RL

In the realm of offline reinforcement learning, the primary objective is to learn effective policies from a static dataset, without the need for online interactions (Levine et al., 2020). In this context, where agents learn from a fixed dataset without interacting with the environment, the exploration aspect is inherently absent. Instead, the focus shifts towards effectively utilizing the available dataset to optimize policies. Traditionally, RL has been concerned with estimating stationary policies or single-step models, leveraging the Markov property to factorize problems in time. However, applying standard RL methods to offline settings is challenging. Because methods relying on value function estimation often suffer from over valuing out-of-distribution states and actions, various methods have been proposed to address this issue, including constraining the policy to be close to the data distribution (Peters et al., 2010), or by using a conservative value function (Kumar et al., 2020).

An intriguing perspective emerges when we view RL through the lens of sequence modeling. Instead of treating it as a specialized domain, we can consider RL as a generic sequence modeling problem. The crux of this viewpoint lies in producing a sequence of actions that leads to a sequence of high rewards. Earlier work has solved this by conditioning the model on returns such that trajectories with high return can be generated in online settings (Ajay et al., 2023; Janner et al., 2021). By adopting this perspective, we can simplify design decisions and dispense with many components commonly found in offline RL algorithms. This approach not only demonstrates flexibility across various tasks such as long-horizon dynamics prediction, imitation learning, goal-conditioned RL, and offline RL but also yields state-of-the-art planners in sparse-reward, long-horizon scenarios (Janner et al., 2022; Ajay et al., 2023).

## 2.3 Denoising Diffusion Probabilistic Models

Denoising diffusion probabilistic models (DDPMs) (Ho et al., 2020) are a type of generative models inspired by non-equilibrium thermodynamics. The forward process slowly add Gaussian noise to data and the reverse amounts to learning to iteratively denoise the noisy data. Diffusion models have primarily been used for image generation, but has also shown state of the art performance on video generation and 3D model generation (Ho et al., 2022; Luo & Hu, 2021).

Given data $\mathbf{x}_0 \sim q(\mathbf{x})$, we define the *forward process* to produce a sequence of noisy samples $\mathbf{x}_1, \ldots, \mathbf{x}_K$,

$$q(\mathbf{x}_k|\mathbf{x}_{k-1}) = \mathcal{N}(\mathbf{x}_k; \sqrt{1 - \beta_k}\mathbf{x}_{k-1}, \beta_k \boldsymbol{I}). \tag{1}$$

where $\{\beta_k \in (0, 1)\}_1^K$ is a carefully chosen variance schedule. A nice property of the forward process is that we can directly sample $\mathbf{x}_k$ at any step $k$. The distribution $q(\mathbf{x}_k|\mathbf{x}_0)$ can be derived using the property that

a sum of uncorrelated normally distributed random variables are normally distributed. Let $a_k = 1 - \beta_k$ and $\overline{a}_k = \prod_{i=1}^{k} a_k$, then

$$q(\mathbf{x}_k|\mathbf{x}_0) = \mathcal{N}\left(\mathbf{x}_k; \sqrt{\overline{a}_k}\mathbf{x}_0, (1 - \overline{a}_k)\,\mathbf{I}\right). \tag{2}$$

Note also that

$$q(\mathbf{x}_{k-1}|\mathbf{x}_k, \mathbf{x}_0) = \mathcal{N}\left(\mathbf{x}_{k-1}; \tilde{\boldsymbol{\mu}}(\mathbf{x}_k, \mathbf{x}_0), \tilde{\beta}_k\boldsymbol{I}\right), \tag{3}$$

where

$$\tilde{\boldsymbol{\mu}}(\mathbf{x}_k, \mathbf{x}_0) = \frac{\sqrt{\overline{a}_{k-1}}\beta_t}{1 - \overline{a}_k}\mathbf{x}_0 + \frac{\sqrt{a_k}(1 - \overline{a}_{k-1})}{1 - \overline{a}_k}, \qquad \tilde{\beta}_k = \frac{1 - \overline{a}_{k-1}}{1 - \overline{a}_k}. \tag{4}$$

While the forward process creates noisy representation of data, the *reverse process* aims to iteratively recreate samples from noise by modelling and then sampling from $q(\mathbf{x}_{k-1}|\mathbf{x}_k)$. Let $p_\theta$ be a parameterized approximation of $q$. This gives

$$p_\theta(\mathbf{x}_{k-1}|\mathbf{x}_k) = \mathcal{N}\left(\mathbf{x}_{k-1}; \boldsymbol{\mu}_\theta(\mathbf{x}_k, k), \Sigma_\theta(\mathbf{x}_k, k)\right). \tag{5}$$

Ho et al. (2020) chose to fix the variance term $\boldsymbol{\Sigma}_\theta(\mathbf{x}_k, k)$ as a constant $\sigma_k^2 = \tilde{\beta}_k$, see Eq (4). Although Nichol & Dhariwal (2021) has shown improved results by learning a parameterization of $\boldsymbol{\Sigma}_\theta(\mathbf{x}_k, k)$, we will only look at how $\boldsymbol{\mu}_\theta(\mathbf{x}_k, k)$ is trained. First, we consider the reparameterization $\tilde{\boldsymbol{\mu}}_\theta = \frac{1}{\sqrt{a_k}}\left(\mathbf{x}_k - \frac{1-a_k}{\sqrt{1-\overline{a}_k}}\boldsymbol{\epsilon}_k\right)$, where $\boldsymbol{\epsilon}_k \sim \mathcal{N}(0, I)$. Since $\mathbf{x}_k$ is known during training, we can choose to predict $\boldsymbol{\epsilon}_k$, rather than $\tilde{\boldsymbol{\mu}}_k$ directly. Empirically, this has shown better results. Let us define $\boldsymbol{\epsilon}_\theta(\mathbf{x}, k)$ as a model that predicts the noise, $\boldsymbol{\epsilon}_k$, added to the input. This means that we can define $\boldsymbol{\mu}_\theta(\mathbf{x}_k, k) = \frac{1}{\sqrt{a_k}}\left(\mathbf{x}_k - \frac{1-a_k}{\sqrt{1-\overline{a}_k}}\boldsymbol{\epsilon}_\theta(\mathbf{x}_k, k)\right)$. Ho et al. (2020) derive the following loss function to minimize the difference between $\boldsymbol{\mu}_\theta$ and $\tilde{\boldsymbol{\mu}}$:

$$L(\theta) = \mathbb{E}_{k\sim[1,K],\mathbf{x}_0,\boldsymbol{\epsilon}_k}\left[\frac{\beta_k^2}{2\sigma_k^2 a_k(1 - \overline{a}_k)}\left\|\boldsymbol{\epsilon}_k - \boldsymbol{\epsilon}_\theta(\mathbf{x}_k, k)\right\|^2\right]. \tag{6}$$

They also present the following simplified loss function that turns out to give better empirical results:

$$L(\theta) = \mathbb{E}_{k\sim[1,K],\mathbf{x}_0,\boldsymbol{\epsilon}_k}\left\|\epsilon_k - \epsilon_\theta(\mathbf{x}_k, k)\right\|^2. \tag{7}$$

## 2.4 Guided Diffusion

There are two main ways diffusion models can condition on variables. One, called classifier-guided diffusion (Dhariwal & Nichol, 2021), uses the gradients of the input to a classifier function, with conditioning classes $y$, with respect to the predicted log-likelihood to alter the noise prediction toward the conditioning information. This method has the advantage that the diffusion model does not have to be trained on the conditioning variable. A model predictor $\overline{\boldsymbol{\epsilon}}_\theta$, guided by a classifier $h(y|\mathbf{x}_k, k)$ meant to estimate the probability that the noisy datapoint $\mathbf{x}_k$ belongs to class $y$ would assume the following form:

$$\overline{\boldsymbol{\epsilon}}_\theta(\mathbf{x}_k, k, y) = \boldsymbol{\epsilon}_\theta(\mathbf{x}_k, k) - w\sigma_k\nabla_{\mathbf{x}_k}\log h(y|\mathbf{x}_k, k), \tag{8}$$

where $w$ is a parameter controlling the strength of the guidance.

The second way, called classifier-free guidance (Ho & Salimans, 2021), plugs the conditioning variable directly into the denoising network as an auxiliary input variable during training. At test time, the auxiliary variable can be set to the conditioning value. In this setting, the model predictor would take the following form:

$$\tilde{\boldsymbol{\epsilon}}(\mathbf{x}_k, k, y) = (w + 1)\boldsymbol{\epsilon}_\theta(\mathbf{x}_k, k, y) - w\boldsymbol{\epsilon}_\theta(\mathbf{x}_k, k) \tag{9}$$

Classifier-free guidance has shown better practical performance than classifier-guided diffusion (Ho & Salimans, 2021).

Beyond the two primary methods of conditioning, we can also employ inpainting (Lugmayr et al., 2022) as a technique to condition on partial observations. In the context of image generation, this implies conditioning on some pixels within the image. Consider an image $\mathbf{x}_0$ divided into known pixels $\mathbf{x}_0^{\text{known}}$ and unknown pixels $\mathbf{x}_0^{\text{unknown}}$, and a mask $\boldsymbol{m}$ defining the known pixels. During the reverse process, we define:

$$\mathbf{x}_{k-1}^{\text{known}} = \sqrt{\overline{a}_k}\mathbf{x}_0 + (1 - \overline{a}_k)\boldsymbol{z}, \qquad \boldsymbol{z} \sim \mathcal{N}(\boldsymbol{0}, I). \tag{10}$$

The unknown pixels at step $k$, $\mathbf{x}_{k-1}^{\text{unknown}}$, are computed in standard fashion:

$$\mathbf{x}_{k-1}^{\text{unknown}} = \frac{1}{\sqrt{a_k}} \left( \mathbf{x}_k - \frac{\beta_k}{\sqrt{1 - \overline{a}_k}} \boldsymbol{\epsilon}_\theta(\mathbf{x}_k, k) \right) + \sigma_k \mathbf{z}, \qquad \mathbf{z} \sim \mathcal{N}(\mathbf{0}, I). \tag{11}$$

Finally, we have:

$$\mathbf{x}_{k-1} = \boldsymbol{m} \odot \mathbf{x}_{k-1}^{\text{known}} + (1 - \boldsymbol{m}) \odot \mathbf{x}_{k-1}^{\text{unknown}}, \tag{12}$$

where $\odot$ is elementwise multiplication. This is optimised for continuous data, and would likely not work well for environments with discrete states. This is due to a multitude of reasons, one being that diffusion models rely on smooth interpolation between states, which discrete data lack.

## 2.5 Bayesian Flow Networks

Bayesian flow networks (Graves et al., 2024) is a novel generative model capable of generating continuous, discrete, and discretized data. It resembles diffusion models in that it generates data in an iterative process. Unlike diffusion models however, it is not a reverse process starting from noisy data, but rather starts from a prior distribution and iteratively updates the distribution conditioned on a noisy version of the previous distribution. Also unlike diffusion models, Bayesian Flow Networks perform well on discrete data, and are therefore a more natural choice for planning in discrete state spaces. Bayesian Flow Networks have been shown to perform well on discretised image and text generation.

A comprehensive description of Bayesian Flow Networks is beyond the scope of this paper, but we aim to give the reader a clear understanding of how they differ from diffusion models. Bayesian Flow Networks (BFNs) are a class of probabilistic models designed to offer a flexible and scalable approach to modeling complex distributions. In BFNs, the inputs to the neural network are parameters of distributions, which remain continuous even for categorical distributions. This characteristic allows BFNs to adapt well to discrete data. A flowchart of the Bayesian flow network algorithm for a single categorical variable is shown in Figure 1.

For a basic description of the discrete case, consider data represented as a $D$ dimensional vector $\mathbf{x} = \left( x^{(1)}, \dots, x^{(D)} \right) \in \{1, A\}^D$, where $A$ is the number of classes, and $\{1, \dots, A\}$ is the set of integers from 1 and $A$. We will model this as a categorical distribution. First we will define the four different distributions, the *input distribution*, the *output distribution*, the *sender distribution*, and the *receiver distribution*, shown in Figure 1.

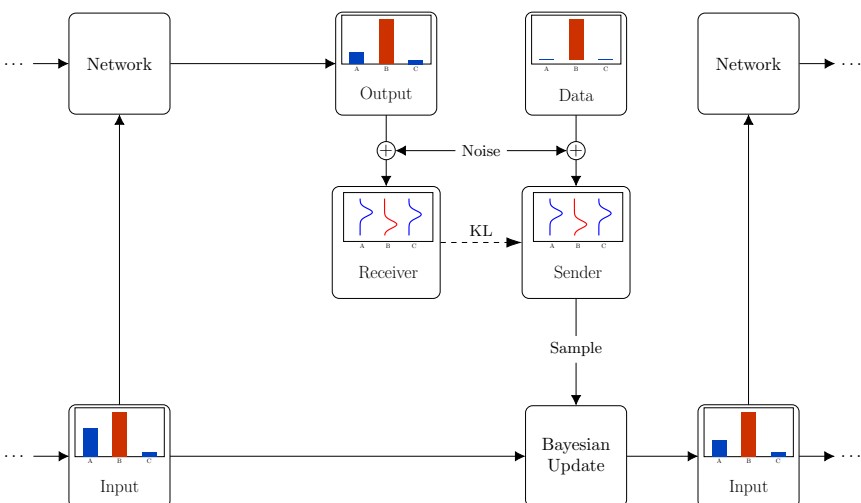

Figure 1: Chart shows one step in the Bayesian flow network process for one categorical variable. Figure is adapted from Figure 1 in Graves et al. (2024).

**Input distribution**   For discrete data, the input distribution is modeled as a factorized categorical with parameters $\boldsymbol{\theta} = \left(\boldsymbol{\theta}^{(1)}, \ldots, \boldsymbol{\theta}^{(D)}\right)$, where each $\boldsymbol{\theta}^{(d)}$ comprises $A$ corresponding to the categorical distribution for variable $d$. Specifically, $\theta_a^{(d)}$ represents the probability assigned to class $a$ for variable $d$.

$$p_I(\mathbf{x}|\boldsymbol{\theta}) = \prod_{d=1}^{D} \boldsymbol{\theta}^{(d)} \tag{13}$$

Initially, the input distribution is uniform, meaning $\boldsymbol{\theta}_0 = \left[\frac{1}{a}, \ldots, \frac{1}{a}\right]$.

**Output distribution**   $\Psi(\boldsymbol{\theta}, k)$ is a neural network model that takes as input a $D$-dimensional parameter vector, $\boldsymbol{\theta}$, where each element are parameters of a categorical distribution. The output is of the same type. The output distribution for discrete data is defined based on the data $\mathbf{x}$, model inputs $\boldsymbol{\theta}, t$, and resulting model outputs $\Psi\left(\boldsymbol{\theta}, k\right) = \left(\Psi^{(1)}\left(\boldsymbol{\theta}, k\right), \ldots, \Psi^{(D)}\left(\boldsymbol{\theta}, k\right)\right) \in \mathbb{R}^{AD}$. The network inputs $\boldsymbol{\theta}$ represent the parameters of the factorized categorical distribution $p_I(\mathbf{x}|\boldsymbol{\theta})$, while $k$ serves as an additional input that represent the process time.

$$p_O(\mathbf{x} \mid \boldsymbol{\theta}, t) = \prod_{d=1}^{D} \Psi^{(d)}(\boldsymbol{\theta}, k) \tag{14}$$

Here, $\Psi^{(d)}(\boldsymbol{\theta}, k)$ denotes $A$ components of the network output corresponding to the parameters $\left(\theta_1^{(d)}, \ldots, \theta_A^{(d)}\right)$ of the categorical distribution for the $d$-th observation.

**Sender distribution**   A sample from the sender distribution is used to update the parameters of the input distribution. The accuracy of these samples is controlled by an accuracy parameter $\alpha \in \mathbb{R}^+$ When $\alpha$ is low, the samples provide limited information about $\mathbf{x}$. As $\alpha$ increases, the samples become increasingly informative about $\mathbf{x}$. For $\mathbf{y} = \left(y^{(1)}, \ldots, y^{(D)}\right) \in \mathcal{Y}^D$, the sender distribution is defined as

$$p_S\left(\mathbf{y} \mid \mathbf{x}; \alpha\right) = \mathcal{N}\left(\mathbf{y} \mid \alpha\left(A\mathbf{e_x} - \mathbf{1}\right), \alpha A \boldsymbol{I}\right), \tag{15}$$

where $\mathbf{e_x}$ is a unit vector of length $A$ and element $\mathbf{x}$ is 1.

**Receiver distribution**   The receiver distribution is defined according to the output distribution $p_O$, and $p_S$, this takes the form

$$p_R(\mathbf{y} \mid \boldsymbol{\theta}; k, \alpha) = \mathbb{E}_{p_O(\mathbf{x}'|\boldsymbol{\theta}; k)}\left[p_S\left(\mathbf{y}|\mathbf{x}'; \alpha\right)\right], \tag{16}$$

In essence, this integrates over all $\mathbf{x}' \in \{1, \ldots, K\}^D$, considering the contribution of each possible $\mathbf{x}'$ as weighted by its likelihood under the output distribution $p_O(\mathbf{x}|\boldsymbol{\theta}, t)$, effectively combines all potential sender distributions into a single receiver distribution.

The objective at each step is to minimize the KL-divergence from the receiver distribution to the sender distribution across all variables. Additionally, after all steps are complete, the objective is to maximize the likelihood of sampling the data from the distribution $p_O$.

We would like a loss function that can be performed at any step $k$ without going through all the previous steps. To do this, we need to know the distribution of the parameters $\boldsymbol{\theta}$ given only the prior $\boldsymbol{\theta}_0$ and the step $k$. Graves et al. (2024) call this the Bayesian flow distribution, $p_F$. This distribution is based on two terms we will introduce now, namely the Bayesian update distribution, and the accuracy schedule.

The Bayesian update function, used to update the input function for each step as shown in Figure 1 is given by

$$h\left(\boldsymbol{\theta}_{i-1}, \mathbf{y}\right) = \frac{e^{\mathbf{y}} \odot \boldsymbol{\theta}_{i-1}}{\sum_{a=1}^{A} e^{\mathbf{y}_a} \left(\boldsymbol{\theta}_{i-1}\right)_a}. \tag{17}$$

This function updates the parameters $\boldsymbol{\theta}_{i-1}$ using new samples $\mathbf{y}$ resulting in $\boldsymbol{\theta}_i \leftarrow h\left(\boldsymbol{\theta}_{i-1}, \mathbf{y}\right)$. We slightly abuse notation and use $e^{\mathbf{y}}$ to mean the elementwise exponentiation of $\mathbf{y}$.

Given a multivariate Dirac delta distribution, $\delta(\cdot)$, the Bayesian update distribution is defined as

$$p_U(\boldsymbol{\theta} \mid \boldsymbol{\theta}_{i-1}, \mathbf{y}, \alpha) = \mathbb{E}_{\mathcal{N}(\mathbf{y}|\alpha(A\mathbf{e}_\mathbf{x}-\mathbf{1}), \alpha A \boldsymbol{I})} \left[\delta\left(\boldsymbol{\theta} - h(\boldsymbol{\theta}_{i-1}, \mathbf{y})\right)\right]. \tag{18}$$

Next, we define the accuracy schedule $\beta(k)$ as the integral of the accuracy rate $\alpha(k)$ over time.

$$\beta(k) = \int_0^k \alpha(k')dk'. \tag{19}$$

In the discrete case, Graves et al. (2024) use the schedule $\beta(k) = \beta(1)k^2$, where $\beta(1)$ is a hyperparameter to be determined empirically for each experiment. Essentially, we calculate the cumulative accuracy up to time $k$, so that we later can perform a single Bayesian update from the prior to time $k$.

Combining the accuracy schedule with the Bayesian update distribution, we obtain the Bayesian flow distribution:

$$p_F(\boldsymbol{\theta} \mid \boldsymbol{\theta}_0, \mathbf{y}, t)) = \mathbb{E}_{\mathcal{N}(\mathbf{y}|\beta(t)(A\mathbf{e}_\mathbf{x}-\mathbf{1}), \beta(t)A\boldsymbol{I})} \left[\delta\left(\boldsymbol{\theta} - h(\boldsymbol{\theta}_0, \mathbf{y})\right)\right]. \tag{20}$$

Graves et al. (2024) define two types of loss functions: the discrete-time loss $L^n$ and the continuous-time loss $L^\infty$. The discrete-time loss $L^n$ corresponds to $n$ generation steps, while $L^\infty$ represents the loss as $n \to \infty$. One advantage of the continuous-time loss is that the number of generation steps can be determined at inference time rather than when training the model. This is the loss we will utilize in our method. Given

$$\hat{\mathbf{e}}^{(d)}(\boldsymbol{\theta}, k) = \sum_{a=1}^{A} p_O(a|\boldsymbol{\theta}; k)\mathbf{e}_a, \tag{21}$$

the continuous time loss is defined as

$$L^\infty(\mathbf{x}) = \mathbb{E}_{k \sim U(0,1), \boldsymbol{\theta} \sim P_F(\cdot|\mathbf{x}, k)} k \left\|\mathbf{e}_\mathbf{x} - \hat{\mathbf{e}}(\boldsymbol{\theta}, k)\right\|^2. \tag{22}$$

This completes the definition of the continuous time loss function for Bayesian flow networks with discrete data.

The inference process begins with initial parameters $\boldsymbol{\theta}_0$, and proceeds through $n$ steps, each characterized by specific accuracies $\alpha_1, \dots, \alpha_n$ and corresponding time points $k_i = \frac{i}{n}$. At each step $i$, the parameters $\boldsymbol{\theta}_i$ are updated recursively as follows:

1. Sample $\mathbf{x}$ from $p_O(\cdot \mid \boldsymbol{\theta}_{i-1}, k_{i-1})$

2. Generate $\mathbf{y}$ from the sender distribution $p_S(\cdot \mid \mathbf{x}, \alpha_i)$.

3. Update the parameters $\boldsymbol{\theta}_i = h(\boldsymbol{\theta}_{i-1}, \mathbf{y})$.

Notice that the sender distribution is now conditioned on the sample from the output distribution and not the data. After completing $n$ steps, with the final parameters $\boldsymbol{\theta}_n$, one last step is performed, and the final sample is drawn from $p_O(\cdot \mid \boldsymbol{\theta}_n, 1)$.

There are no obvious factors stopping Bayesian Flow Networks from utilizing the same conditioning techniques as diffusion models, but the authors are not aware of any implementations of inpainting (Lugmayr et al., 2022) or classifier guidance at the time of writing. Graves et al. (2024) specifically state that BFNs pave the way for gradient-based sample guidance in discrete domains, however.

## 3 Related Work

Offline RL as a sequence modeling problem has been explored in several recent works.

### 3.1 Diffuser

The Diffuser model (Janner et al., 2022) uses an unconditional diffusion model to model state-action sequences. Since this model is unconditional, a differentiable reward function trained on noisy state-action pairs is necessary to guide the model. A u-net architecture with 1-dimensional local receptive field is used to model the diffusion process. With this architecture, the model is able to model sequences of arbitrary length, and can be conditioned on a desired return at test time. Additionally, the local receptive fields makes the model learn local consistency that can evolve into global consistency through many diffusion steps. At test-time, the model can be conditioned on the current state by using inpainting, and the desired return by using classifier-free guidance. They further show that the inpainting technique can be used to condition on desired end states, effectively allowing the model to solve planning problems it was not specifically trained for.

### 3.2 Decision Diffuser

The Decision Diffuser Ajay et al. (2023) is similar to the Diffuser method, but differs in two main ways; how it models actions, and how it conditions on rewards. First, the Decision Diffuser leverages an inverse dynamics model to capture the relationship between states and actions. This inverse dynamics model estimates actions conditioned on states, effectively predicting the action that brought the environment from one state to the next. This lets the diffusion model only model the state sequences rather than state-action sequences. They show empirically that using an inverse dynamics model is advantageous in deterministic environments, but that as more stochasticity is introduced into the environment the performance reduces to the same level as the Diffuser. The second way in which the Decision Diffuser differs from the Diffuser is that it conditions on return-to-go in a classifier free manner. What this means is that the return is fed into the model during training so that the model learns which sequences to associate with that return. The desired return can again be fed into the model when generating a sequence of future states.

## 4 Method

We propose a sequence generating approach to reinforcement learning based on Bayesian flow networks capable of planning in both discrete and continuous domains. From now on we will refer to our method as BFN-RL. Like Decision Diffuser (Ajay et al., 2023) we will only model the state sequences conditioned on current state and future return and utilize a second inverse-dynamics network to model the actions conditioned on the states. We opt for this method, as it showed superior performance compared to modeling state-action pairs with a single diffusion model (Janner et al., 2022). We expect the same benefits when using Bayesian flow networks as the generative model, but future work may reveal differently.

In our method, the Bayesian flow network is trained to generate sequences conditioned on the return and current state. The return is the sum of all discounted future rewards, and is therefore a measure of the quality of a sequence. The network learns to model the distribution of sequences with both high and low return, and can at test time be conditioned on a desired return. The network is not specifically trained to generate sequences with high return, but rather at test-time it can be conditioned on a high return value and perform plans that outperform any seen in the dataset.

## 4.1 Condition on Return

There are two obvious ways we can condition on return, the Diffuser (Janner et al., 2022) way, or the Decision Diffuser (Ajay et al., 2023) way. Considering that the Decision Diffuser is significantly easier to implement, showed better performance, and does not involve training an extra model, we opted to condition directly on return in a classifier-free manner as was done in Decision Diffuser (Ajay et al., 2023). This is shown in Figure 2, where the neural network in each BFN-step gets the current parameters for the state-distribution factorized over each timestep, as well as the return and the step $k$. Int the figure, $s_t$ refers to the state of the environment at time $t$, $\hat{s}_{t+1,k}$ refers to the noisy state estimate at time $t+1$ after $k$ BFN steps.

By directly conditioning on return during the generation process, we ensure that the generated trajectories are biased towards high-reward regions of the state-action space, promoting the discovery of effective policies that maximize cumulative rewards. Moreover, leveraging the Decision Diffuser framework allows for seamless integration with existing generative modeling architectures, such as the temporal u-net architecture used in Decision Diffuser (Ajay et al., 2023).

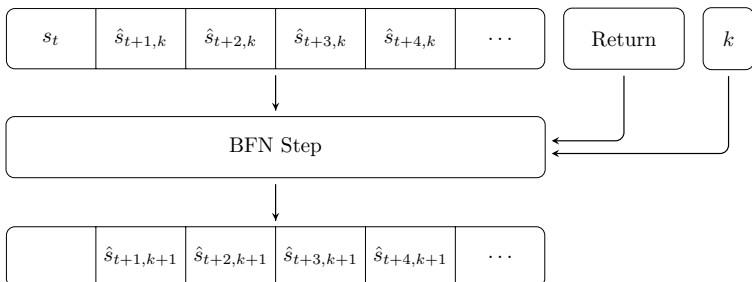

Figure 2: Each BFN step is conditioned on $s_t$, return, and step $k$ to generate a sequence of future states.

## 4.2 Conditioning on Current State

When Diffuser (Janner et al., 2022) and Decision Diffuser (Ajay et al., 2023) conditions on the current state, they apply an inpainting technique specific to diffusion models. During the reverse diffusion process, the part of the sequence that is known, the first state, is at each timestep replaced by the true value diffused the appropriate amount for that timestep. We have adapted a similar technique for Bayesian flow networks. For continuous data, the Bayesian update at each step for conditional variable is made in the correct direction. Similarly, for discrete data, the probabilities of the categorical distribution is set to the appropriate probability for that step. Algorithm 1 shows this method implemented for discrete data. The alterations to BFN sampling (Algorithm 9 in Graves et al. (2024)) are highlighted in red.

A different way of conditioning on current state is to implement this directly into the model in the same way we condition on return. We call this method direct conditioning. When using direct conditioning, the model must be given conditioning variables also at training time. Rather than modify the network to accept a separate input for the condition, we choose to let the first step in the sequence denote the conditioning variable. This has the advantage that the conditioning variable will be more directly changing the beginning of the sequence than the end. Algorithm 2 shows how discrete conditioning would change the sampling algorithm for discrete variables. The mask, $\boldsymbol{m}$, masks out the first step in the sequence so that we condition on this step rather than generate it. The condition, $\boldsymbol{c}$, is the current state. The red lines indicate lines added on top of the regular BFN algorithm. In Algorithm 1 a mask $\boldsymbol{m}$ indicates which variables should be sampled from the sender distribution, and which should be sampled from the data. Algorithm 2 on the other hand, changes the input distribution such that the data is used directly for the conditioned variable. Preliminary experiments showed that this gave significantly better results for discrete variables, and similar or slightly better results for continuous experiments. Based on this, we proceed to use the method presented in Algorithm 2.

---

**Algorithm 1** Inpainting Conditioning for Discrete Random Variables

---

**Require:** $\beta(1) \in \mathbb{R}^+$, number of steps $n \in \mathbb{N}$, number of classes $K$, mask $\boldsymbol{m}$, condition $\boldsymbol{c}$

    $\boldsymbol{\theta} \leftarrow \frac{1}{K}$
    **for** $i = 1$ to $n$ **do**
        $t \leftarrow \frac{i-1}{n}$
        $\mathbf{k} \sim \text{DISCRETE\_OUTPUT\_DISTRIBUTION}(\boldsymbol{\theta}, t)$
        $\alpha \leftarrow \beta(1) \left(\frac{2i-1}{n^2}\right)$
        $\mathbf{y} \sim \mathcal{N}\left(\alpha(K\mathbf{e_k} - \mathbf{1}), \alpha K \boldsymbol{I}\right)$
        $\mathbf{y_c} \leftarrow \alpha(K\mathbf{e_c} - \mathbf{1})$
        $\mathbf{y} \leftarrow (\mathbf{1} - \boldsymbol{m}) \odot \mathbf{y_c} + \boldsymbol{m} \odot \mathbf{y}$
        $\boldsymbol{\theta}' \leftarrow e^{\mathbf{y}}\boldsymbol{\theta}$
        $\boldsymbol{\theta} = \frac{\boldsymbol{\theta}'}{\sum_k \boldsymbol{\theta}'_k}$
    **end for**
    $\mathbf{k} \sim \text{DISCRETE\_OUTPUT\_DISTRIBUTION}(\boldsymbol{\theta}, 1)$

---

**Algorithm 2** Direct Conditioning for Discrete Random Variables

---

**Require:** $\beta(1) \in \mathbb{R}^+$, number of steps $n \in \mathbb{N}$, number of classes $K$, mask $\boldsymbol{m}$, condition $\boldsymbol{c}$

    $\boldsymbol{\theta} \leftarrow \frac{1}{K}$
    **for** $i = 1$ to $n$ **do**
        $t \leftarrow \frac{i-1}{n}$
        $\boldsymbol{\theta} \leftarrow (\mathbf{1} - \boldsymbol{m}) \odot \mathbf{e_c} + \boldsymbol{m} \odot \boldsymbol{\theta}$
        $\mathbf{k} \sim \text{DISCRETE\_OUTPUT\_DISTRIBUTION}(\boldsymbol{\theta}, t)$
        $\alpha \leftarrow \beta(1) \left(\frac{2i-1}{n^2}\right)$
        $\mathbf{y} \sim \mathcal{N}\left(\alpha(K\mathbf{e_k} - \mathbf{1}), \alpha K \boldsymbol{I}\right)$
        $\boldsymbol{\theta}' \leftarrow e^{\mathbf{y}}\boldsymbol{\theta}$
        $\boldsymbol{\theta} = \frac{\boldsymbol{\theta}'}{\sum_k \boldsymbol{\theta}'_k}$
    **end for**
    $\boldsymbol{\theta} \leftarrow (\mathbf{1} - \boldsymbol{m}) \odot \mathbf{e_c} + \boldsymbol{m} \odot \boldsymbol{\theta}$
    $\mathbf{k} \sim \text{DISCRETE\_OUTPUT\_DISTRIBUTION}(\boldsymbol{\theta}, 1)$

---

### 4.3 Inverse Dynamics Model

Just like Decision Diffuser (Ajay et al., 2023), we opt to only model the state sequence with the generative model, and use an inverse dynamics model to predict the action that takes the environment from one state to the next. Preliminary experiments showed that this method performed better than generating state-action pairs also with Bayesian flow networks as the generative model.

## 5 Experiments

We evaluate our method on two sets of tasks, one with discrete action and state space, and one with continuous action and state space. In the discrete case we use a gridworld environment. For the continuous case we use the D4RL (Fu et al., 2020) datasets with the Gym-Mujoco suite of environments. We compare our method to the Decision Diffuser (Ajay et al., 2023) and other state of the art offline RL methods.

### 5.1 Gridworld

Our discrete environment experiments look at the method's ability to plan in both stochastic and deterministic environments. The first environment, SingleRoomUndirected, is a simple grid world with a player and a goal in a $6 \times 6$ grid surrounded by walls. There are 4 actions that can take the player up, down, left, or right.

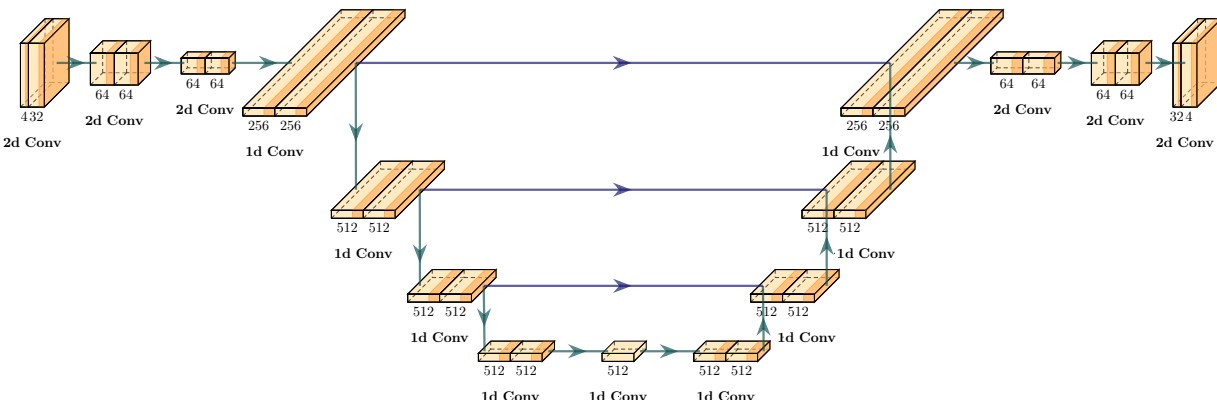

Figure 3: Specialized u-net architecture used for GridWorld problems. A series of 2d convolutions is performed before the 1d temporal u-net, and another series of 2d transposed convolutions are performed after the tmeporal u-net. This ensures that the model can take advantage of the 2d structure of the GridWorld problems and generalize better to unseen sequences.

This environment serves as a testbed for evaluating the method's performance in a controlled setting, allowing us to assess its ability to navigate and reach the goal efficiently. Additionally, we introduce variations of this environment to explore the method's robustness to stochasticity. This comparative analysis provides valuable insights into the method's reliability and effectiveness in discrete state-space environments.

DynamicObstaclesUndirected is a different environment, but also with a 6x6 grid surrounded by walls and a goal square. Additionally, it has a number of obstacles that moves around the grid taking by randomly going up, down, left, or right for each timestep. We test the agent against 1, 2, or 4 obstacles to see how it performs in an increasing stochastic environment. A score of 1 is given if the agent reaches the goals state, a score of -1 if it hits any obstacles, and a score of 0 if it terminates after 32 steps without reaching the goal or hitting an obstacle. The dataset is collected by a random agent over 100k episodes.

We propose a novel neural network architecture tailored specifically for gridworld environments. Leveraging the inherent structure of gridworlds, our architecture starts by applying 2D convolutional operations to each frame of sequence. This approach allows the network to capture spatial relationships within the grid. Subsequently, the feature maps are flattened and fed into a temporal u-net, enabling the network to learn temporal dependencies across frames. This design choice facilitates generalization to unseen transitions within the gridworld while at the same time mitigates memory consumption compared to a 3D convolutional approach. The temporal u-net part is similar to the one used in Diffuser and Decision Diffuser, and consists of temporal convolutions, group normalization, and Mish activation functions.

In the discrete case we opt to use an analytic inverse dynamics model. Given two subsequent states, we look at what direction the player moved in. If no action can move the player to the new state, a new plan is generated from the current state. This assumes that player movement is deterministic.

Table 1 shows the methods ability to create valid state sequences in a simple deterministic environment. We observe that as the number of BFN steps increases, a larger portion of the generated state sequences become valid. A state sequence is considered valid if there exists a policy $\pi$ such that following this policy from the conditioned start state may generate the given state sequence. Furthermore, we observe that all the valid state-sequences that were generated were of the desired length.

Table 2 shows the performance of BFN-RL DynamicObstaclesUndirected with 0, 1, 2, and 4 dynamic obstacles. Surprisingly, the results for 4 obstacles are significantly better than for 2 obstacles. In general though, the results suggest that our method struggles with more stochastic domains. Another possibility for the reduced performance with more obstacles is that the model did not have the capacity to capture the increased complexity of the environment.

| Steps | Fraction of plans that are valid | Fraction of valid plans that produce the correct score |
|---|---|---|
| 1 | 0.01 | 1.00 |
| 5 | 0.46 | 1.00 |
| 10 | 0.70 | 1.00 |
| 100 | 0.74 | 1.00 |

Table 1: This table shows how many plans in the discrete environment SingleRoomUndirected were valid for the entire horizon, and how many of those gave the correct score.

| Dynamic Obstacles | Random | BFN-RL |
|---|---|---|
| 0 | 0.36 | 1.00 |
| 1 | $-0.06$ | 0.90 |
| 2 | $-0.34$ | $-0.05$ |
| 4 | $-0.60$ | 0.45 |

Table 2: This table shows the return on the DynamicObstaclesUndirected environment with a varying number of obstacles. Random refers to the expected return of a random agent calculated from 10k simulations. Returns for BFN-RL are averaged over 20 seeds.

## 5.2 Continuous Control

While the focus of our method lies in addressing challenges in discrete state-spaces, something Diffuser(Janner et al., 2022) and Decision Diffuser (Ajay et al., 2023) cannot easily do, we also aim to demonstrate its effectiveness in continuous settings. To this end, we evaluate our method on the D4RL MuJoCo dataset. By extending our method to continuous environments, we aim to showcase its adaptability and versatility across different problem domains. The D4RL MuJoCo dataset provides a comprehensive benchmark for evaluating algorithms in continuous control tasks, offering a diverse range of simulated environments with varying complexities. Through these evaluations, we aim to demonstrate that our method is not limited to discrete settings but can also effectively handle continuous environments.

Table 3 shows the performance of BFN-RL compared to state-of-the-art algorithms. The table shows that BFN-RL is competitive on most datasets, but clearly struggles on the hopper environment. The method is generally very sensitive to hyperparameters, and we believe that results on the Hopper datasets could likely be improved with further tuning specific to Hopper. HalfCheetah and Walker2d have the same 17-dimensional input space, whereas Hopper has an 11-dimensional input space. We hypothesize that this discrepancy, or something related to the dynamics of the agent, could require different hyperparameters for BFN to effectively learn the environment dynamics. To make results with Decision Diffuser comparable, we opted not to specifically tune hyperparameters for each environment.

## 6 Conclusion

In this work, we introduced a novel approach to reinforcement learning that leverages Bayesian flow networks (Graves et al., 2024) for sequence generation. Our method is capable of planning in both discrete and continuous domains by conditioning on returns and current states. We explored two strategies for conditioning: conditioning on the current state using inpainting as seen in the Decision Diffuser (Ajay et al., 2023), and conditioning on current state in a classifier-free manner. Our framework simplifies the training pipeline and reduces computational overhead by eliminating the need for an additional return classifier.

Our experiments demonstrated the effectiveness of our approach across various environments. In the discrete setting, our method successfully navigated gridworld environments, showcasing its ability to generate valid plans and achieve desired outcomes. The specialized neural network architecture, which combines 2D convolutions with a temporal u-net, effectively captured spatial and temporal dependencies, leading to robust performance. In continuous environments, our method's adaptability was evident from its competitive performance on the D4RL MuJoCo datasets, highlighting its versatility across different problem domains.

| Dataset | Environment | BC | CQL | IQL | DT | TT | MOReL | Diffuser | DD | BFN-RL |
|---|---|---|---|---|---|---|---|---|---|---|
| Med-Expert | HalfCheetah | 55.2 | 91.6 | 86.7 | 86.8 | **95** | 53.3 | 79.8 | 90.6 | 93.4 ± 1.3 |
| Med-Expert | Hopper | 52.5 | 105.4 | 91.5 | 107.6 | **110.0** | 108.7 | 107.2 | **111.8** | 93.4 ± 3.0 |
| Med-Expert | Walker2d | **107.5** | 108.8 | 109.6 | 108.1 | 101.9 | 95.6 | 108.4 | 108.8 | 106.6 ± 0.2 |
| Medium | HalfCheetah | 42.6 | 44.0 | 47.4 | 42.6 | 46.9 | 42.1 | 44.2 | **49.1** | 45.8 ± 0.4 |
| Medium | Hopper | 52.9 | 58.5 | 66.3 | 67.6 | 61.1 | **95.4** | 58.5 | 79.3 | 45.1 ± 1.8 |
| Medium | Walker2d | 75.3 | 72.5 | 78.3 | 74.0 | 79 | 77.8 | 79.7 | **82.5** | 75.3 ± 2.3 |
| Med-Replay | HalfCheetah | 36.6 | **45.5** | **44.2** | 36.6 | 41.9 | 40.2 | 42.2 | 39.3 | 35.6 ± 1.3 |
| Med-Replay | Hopper | 18.1 | 95 | 94.7 | 82.7 | 91.5 | 93.6 | 96.8 | **100** | 42.1 ± 3.4 |
| Med-Replay | Walker2d | 26.0 | 77.2 | 73.9 | 66.6 | **82.6** | 49.8 | 61.2 | 75 | 54.7 ± 2.8 |
| **Average** | | 51.9 | 77.6 | 77 | 74.7 | 78.9 | 72.9 | 75.3 | **81.8** | 65.8 |

Table 3: **Offline Reinforcement Learning.** The table summarizes the test performance of BFN-RL and a various other methods on continuous control. The results indicate that BFN-RL can match. We report mean and standard error over 3 random seeds. All numbers except for BFN-RL are taken from Ajay et al. (2023)

The results from both discrete and continuous tasks indicate that our approach can generalize well to diverse reinforcement learning challenges.

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

## Appendix

## Hyperparameters

Here we present hyperparameters used in the experiments. For the discrete experiments, we used:

- $\beta(1) = 3$.

- Planning horizon $H = 32$ for SingleRoomUndirected and $H = 4$ for DynamicObstaclesUndirected.

- Learning rate $2e - 4$, adam optimizer (Kingma & Ba, 2014) with $(\beta_1, \beta_2) = (0.9, 0.999)$

- $K = 100$ steps were used for sample generation.

Most hyperparameters and model architectures for continuous experiments that are not specific to Bayesian Flow Networks are similar to those used in the official Decision Diffuser implementation. Hyperparameters used for the continuous experiments:

- Inverse dynamics model is an MLP with two layers with 512 units and ReLU activations.

- $\epsilon_\theta$ and $f_\phi$ are trained for $2e6$ steps using the Adam optimiser (Kingma & Ba, 2014) with a batch size of 64, a learning rate of $2e-4$, and $(\beta_1, \beta_2) = (0.9, 0.98)$.

- We use a planning horizon $H$ of 20.

- For testing we used an exponential moving average of the weights with decay $\alpha = 0.999$

- $K = 200$ steps were used for sample generation.

- $\sigma_1 = 0.01$.

## Algorithms

Algorithm 3 and 4 show an implementation of Algorithm 1 and 2 for continuous data.

---

**Algorithm 3** Inpainting Conditioning for Continuous Random Variables

---

**Require:** $\sigma_1 \in \mathbb{R}^+$, number of steps $n \in \mathbb{N}$, mask $\boldsymbol{m}$, condition $\boldsymbol{c}$
  $\boldsymbol{\mu} \leftarrow \mathbf{0}$
  $\rho \leftarrow 0$
  **for** $i = 1$ to $n$ **do**
    $t \leftarrow \frac{i-1}{n}$
    $\hat{\mathbf{x}}(\boldsymbol{\theta}, t) \leftarrow \text{CTS\_OUTPUT\_DISTRIBUTION}(\boldsymbol{\mu}, t, 1 - \sigma_1^2)$
    $\alpha \leftarrow \sigma_1^{-2i/n}\left(1 - \sigma_1^{2/n}\right)$
    $\mathbf{y} \sim \mathcal{N}\left(\hat{\mathbf{x}}(\boldsymbol{\theta}, t), \alpha^{-1}\boldsymbol{I}\right)$
    $\mathbf{y_c} \leftarrow (1 - \sigma_1^{2t})\boldsymbol{c}$
    $\mathbf{y} \leftarrow (\mathbf{1} - \boldsymbol{m}) \odot \mathbf{y_c} + \boldsymbol{m} \odot \mathbf{y}$
    $\boldsymbol{\mu} \leftarrow \frac{\rho\boldsymbol{\mu} + \alpha\mathbf{y}}{\rho + \alpha}$
    $\rho \leftarrow \rho + \alpha$
  **end for**
  $\hat{\boldsymbol{x}}(\boldsymbol{\theta}, 1) \leftarrow \text{CTS\_OUTPUT\_DISTRIBUTION}(\boldsymbol{\mu}, 1, 1 - \sigma_1^2)$

---

**Algorithm 4** Direct Conditioning for Continuous Random Variables

---

**Require:** $\sigma_1 \in \mathbb{R}^+$, number of steps $n \in \mathbb{N}$, mask $\boldsymbol{m}$, condition $\boldsymbol{c}$
  $\boldsymbol{\mu} \leftarrow \mathbf{0}$
  $\rho \leftarrow 0$
  **for** $i = 1$ to $n$ **do**
    $t \leftarrow \frac{i-1}{n}$
    $\hat{\mathbf{x}}(\boldsymbol{\theta}, t) \leftarrow \text{CTS\_OUTPUT\_DISTRIBUTION}(\boldsymbol{\mu}, t, 1 - \sigma_1^2)$
    $\alpha \leftarrow \sigma_1^{-2i/n}\left(1 - \sigma_1^{2/n}\right)$
    $\mathbf{y} \sim \mathcal{N}\left(\hat{\mathbf{x}}(\boldsymbol{\theta}, t), \alpha^{-1}\boldsymbol{I}\right)$
    $\boldsymbol{\theta} \leftarrow (\mathbf{1} - \boldsymbol{m}) \odot \mathbf{c} + \boldsymbol{m} \odot \boldsymbol{\theta}$
    $\boldsymbol{\mu} \leftarrow \frac{\rho\boldsymbol{\mu} + \alpha\mathbf{y}}{\rho + \alpha}$
    $\rho \leftarrow \rho + \alpha$
  **end for**
  $\boldsymbol{\theta} \leftarrow (\mathbf{1} - \boldsymbol{m}) \odot \mathbf{c} + \boldsymbol{m} \odot \boldsymbol{\theta}$
  $\hat{\boldsymbol{x}}(\boldsymbol{\theta}, 1) \leftarrow \text{CTS\_OUTPUT\_DISTRIBUTION}(\boldsymbol{\mu}, 1, 1 - \sigma_1^2)$

---

## Discrete Experiments

Figure 4 show two plans generated in the SingleRoomUndirected environment, one with 10 steps, and one with 16 steps.

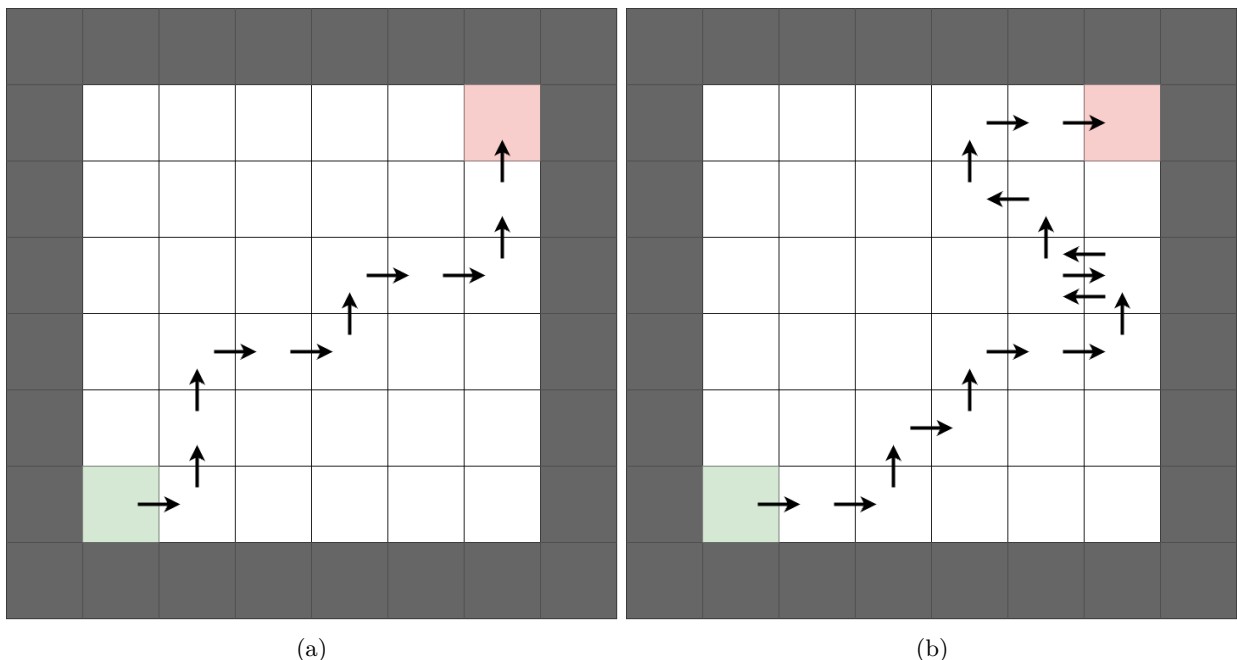

(a)                                                                    (b)

Figure 4: Plans generated by BFN-RL in SingleRoomUndirected. (a) is conditioned to generate a plan of length 10, and (b) is conditioned to generate a plan of length 16.

