# OpenReview forum: "Oﬄine Reinforcement Learning with Bayesian Flow Networks"
_TMLR — Withdrawn by Authors_

### Review · Reviewer_3QMb · 2024-07-27

**Summary Of Contributions:**

In this paper, the authors use Bayesian flow networks (BFNs) to model state sequences in reinforcement learning. This work builds on existing work on using generative models to learn state sequences in RL. The authors introduce methods to condition on returns and current states in BFNs. One advantage of BFNs over diffusion models is that it unifies modeling continuous and discrete distributions. The authors provide empirical evaluations on two class of environments: a gridworld environment with discrete state and action space, and continuous control with continuous state and action space.

**Audience:**

Yes

**Broader Impact Concerns:**

No broader impact concerns.

**Claims And Evidence:**

No

**Requested Changes:**

1.  I appreciate the authors thoroughly explaining the necessary backgrounds. However, the exposition on diffusion models takes up a lot of space. I think devoting more discussion to BFNs and their advantages over other generative models would be more beneficial to the readers.

2. One motivation of conditioning on rewards is that at test time, one could obtain state-action sequences with higher rewards than those in the training dataset. Can the authors discuss to what degree this is achieved by BFN-RL?

3. Some minor change requests:

On page 5 the paragraph on input distribution, $\theta_0$ should have $\frac{1}{A}$ instead of $\frac{1}{a}$ as its entries.

In the paragraph on sender distribution, what is $Y$?

In Table 2, please add the standard deviation to the returns.

What model architecture did the authors use for the continuous control environment? Is it still image-based as in the gridworld one?

In Table 3, please explain what these different dataset names are.

In the baseline comparisons in Table 3, did all the methods use the same amount of offline data? This is not clear from the text.

**Strengths And Weaknesses:**

Strengths:
1. Tacking RL from sequence modeling is an interesting area of research. This work introduces a new approach, which is using BFNs. It is good to study quality differences brought about by using different modeling frameworks.

2. The authors introduce novel ways to condition on rewards and current states in BFNs. This is an important first step to allow control in sequence generation and opens the door to further research.

Weaknesses:
1. The experiment section needs to be improved in both quantity and quality. The gridworld results are inconclusive.  In the DynamicObstaclesUndirected environment, the authors mentioned the degradation with increasing the number of obstacles. It would be good to see some ablation studies on what factors in the design are promising in overcoming this issue. Would more offline data help? How about increasing the number of steps? I believe adding these ablation studies would strengthen this submission and provide valuable information to people who would like to build on this work.

2. The continuous control experiments show that BFN-RL approach is not yet competitive with diffusion-based approaches. It begs the question of why one would choose BFN over diffusion. The authors should provide more discussion on the advantage of BFN over diffusion. A unified representation is one. Are there others, for example, do BFNs enjoy faster inference compared to diffusion models?

---

### Review · Reviewer_B6AC · 2024-07-31

**Summary Of Contributions:**

The rigor and presentation needs serious work

According to the abstract and introduction, the paper studies online RL, where a conditional generative model is trained to output a policy conditioned on return or goal state. The generative model of choice here is the Bayesian Flow Network.

**Audience:**

Yes

**Broader Impact Concerns:**

The broader impact will be evaluated when the authors provide a polished draft.

**Claims And Evidence:**

No

**Requested Changes:**

I mention a few here, and the rest are the authors' responsibility to consult with an ML expert in their writing.

Minor issues:

- When defining an acronym, please use it.
- The authors state, "THE goal of RL is learning a policy .... that maximizes ... "
Please note that RL is a field of study, and in RL, we are concerned with many problem settings, one of which is about finding optimal policy in infinite horizon discounted reward. That is not THE goal.
- "Traditionally, RL has been concerned with estimating stationary policies or single-step models." That is incorrect. Many researchers study non-stationary Markovian policies in offline RL. I assume by the single-step model; the authors mean model-based RL with Markovian assumption, not that the policy model is single-step myopic policy.

-"Also unlike diffusion models, Bayesian Flow Networks perform well on discrete data, and are therefore a more natural choice for planning in discrete state spaces." there are variants of diffusion modes on discrete data that perform well. E.g., "https://arxiv.org/abs/2310.16834"



Major issues:
The section 2.5 is very hard to follow.

"conditioned on a noisy version of the previous distribution." What is the previous distribution?

The {1,A} in {1,A}^D means a set of two members. Why use A? why not just 1 and 2?

In Eq 13, there is no x on the RHS, while there is x on the LHS.

"Initially, the input distribution is uniform, meaning θ0 = 󰀅 a1, . . . , a1 󰀆." Why is that?

The neural network \psi seems to take theta and k as inputs, but the text says, "is a neural network model that takes as input a D-dimensional parameter vector, θ, where each element are parameters of a categorical distribution". It is not contradictory. Also, what is k?

"The output is of the same type." Please be more precise; what does it mean to be the same type?

Above Eq14, it says that \Psi(theta, k) is a vector of dimension AD. However, the notation used is standard of A\times D dimensional matrix.

"Thenetworkinputsθrepresent the parameters of the factorized categorical distribution pI(x|θ), while k serves as an additional input that represent the process time." What is process time?

In Eq 14, there is no t on the RHS, and there is t on the LHS. There is also no x on the RHS, where there is x on the LHS. Please be more rigorous.

"The accuracy of these samples is controlled by an accuracy parameter " what is accuracy?

"where ex is a unit vector of length A and element x is 1." I am not sure I am following this sentence. What does it mean "and element x is 1"?

"We would like a loss function that can be performed at any step k without going through all the previous steps." It is not clear what is a step to talk about the previous and lose it.

To this point, I could not read the paper further since the negotiation is very problematic. I encourage the authors to seriously consider making the writing of this work sound.

**Strengths And Weaknesses:**

The framework is interesting and sounds promising.  The idea is sound, and the authors mentioned a positive empirical study.

However, the presentation of the paper could be stronger and the authors are highly encouraged to consult the writing of their paper seriously.

---

### Review · Reviewer_t1rU · 2024-08-10

**Summary Of Contributions:**

The paper studies the application of Bayesian Flow Networks (BFN) in the context of Offline RL setup. BFN is trained offline to generate trajectories of future states conditioned on the current state and the return. Results for are reported for the following environments: two versions of a 6x6 grid world and varying difficulty datasets for HalfCheetah, Hopper, and Walker2d from D4RL dataset.

**Audience:**

Yes

**Claims And Evidence:**

No

**Requested Changes:**

See the above discussion.

**Strengths And Weaknesses:**

Strengths:
* The current and future generation of generative models can be a game-changer for RL by bringing entirely new capabilities: world models, oracles, reward shaping, code as policies, etc. (say skill reuse, knowledge, exploration, knowledge retrieval, etc.). Consequently, studying the use of novel generative AI methods in the RL context is an important research endeavor.


Weaknesses:
* The paper is unfocused:
	* Large parts of the paper are dedicated to relatively unrelated topics (pages 2-6).
	* Missed opportunity to clearly explain how BFN works and what is the role of each component, particularly in the situation considered in the paper (i.e., RL and the fact that the vectors $x$ are states in RL trajectories).
		* There are also errors in the formulas (e.g., the last paragraph on page 4 should state $\\{1,\ldots, A\\}^D$, RHS of (13) or (14) do not include $x$, $\theta_0$ is hard to interpret: what is $a$? why is it not in $\mathbb R^{AD}$? def of $e_x$ is unclear, etc.).
	* The actual method is poorly described:
		* Perhaps the most informative part here is the last paragraph of page 7.
		* There is no description of the training objective, no pseudo-code for training, and no pseudo-code for inference.
		* Alg 1 and 2 are hard to connect to the actual RL setup in the paper. For starters, there is no mention of states or trajectories, and the notation in Figure 2 does not match.
		* There is no clear discussion of the training data and the underlying assumption. For instance, for the grid worlds, it should be stated that the underlying assumption is that the random policy sufficiently explores the state space, which is quite strong and does not hold for many important environments.
* Experiments are poorly described
	* What are the parameters of BFN?
	* What are the rewards used to condition?
	* What are the insights into the trajectories? How much do they cover the state space? Can we plot the heat maps?
	* How do the various methods compare regarding the computational budget used ($k$, training size, model size, etc.)?
        * What do the particular components of BFN contribute to? What insights can we gain?
* Grid worlds are very toy environments.
* Results are unconvincing.

Other:
* It seems that the paper should cite [1].

[1] Schmidhuber, J., Reinforcement Learning Upside Down: Don’t Predict Rewards - Just Map Them to Actions, 2020

---

### Note · Authors · 2024-08-20

I have read and agree with the venue's withdrawal policy on behalf of myself and my co-authors.